# Optimal High-Dimensional Entanglement Concentration for Pure Bipartite Systems

**DOI:** 10.3390/mi14061207

**Published:** 2023-06-07

**Authors:** Lukas Palma Torres, Miguel Ángel Solís-Prosser, Omar Jiménez, Esteban S. Gómez, Aldo Delgado

**Affiliations:** 1Departamento de Ciencias Físicas, Facultad de Ingeniería y Ciencias, Universidad de La Frontera, Avenida Francisco Salazar 01145, Temuco 4811230, Chile; l.palma03@ufromail.cl; 2Centro de Óptica e Información Cuántica, Facultad de Ciencias, Universidad Mayor, Camino La Pirámide 5750, Huechuraba, Santiago 8580745, Chile; omar.jimenez@umayor.cl; 3Departamento de Física, Universidad de Concepción, Casilla 160-C, Concepción 4070043, Chile; estesepulveda@udec.cl (E.S.G.); aldelgado@udec.cl (A.D.); 4Millennium Institute for Research in Optics, Universidad de Concepción, Casilla 160-C, Concepción 4070043, Chile

**Keywords:** entanglement concentration, Schmidt number

## Abstract

Considering pure quantum states, entanglement concentration is the procedure where, from *N* copies of a partially entangled state, a single state with higher entanglement can be obtained. Obtaining a maximally entangled state is possible for N=1. However, the associated success probability can be extremely low when increasing the system’s dimensionality. In this work, we study two methods to achieve a probabilistic entanglement concentration for bipartite quantum systems with a large dimensionality for N=1, regarding a reasonably good probability of success at the expense of having a non-maximal entanglement. Firstly, we define an efficiency function Q considering a tradeoff between the amount of entanglement (quantified by the I-Concurrence) of the final state after the concentration procedure and its success probability, which leads to solving a quadratic optimization problem. We found an analytical solution, ensuring that an optimal scheme for entanglement concentration can always be found in terms of Q. Finally, a second method was explored, which is based on fixing the success probability and searching for the maximum amount of entanglement attainable. Both ways resemble the Procrustean method applied to a subset of the most significant Schmidt coefficients but obtaining non-maximally entangled states.

## 1. Introduction

Quantum entanglement is the most known, remarkable, and useful quantum resource in the quantum information (QI) theory [1] as it underlies several QI protocols, such as dense coding [2], entanglement swapping [3], quantum teleportation [4], and quantum cryptography [5]. For instance, in the bipartite scenario, two users who want to communicate—usually called Alice and Bob—can share an entangled state [6]. In this case, the ability to transmit information encoded in the state shared by Alice and Bob depends on the amount of entanglement [7,8]. Moreover, the most favorable case for faithful communication is when Alice and Bob share a maximally pure entangled state (MES) [9]. However, even if it was the initial state, the quantum noisy channel used to send the information will produce a loss of correlations in the MES [10]. Moreover, the quantum operations needed to carry out a particular quantum application are performed imperfectly due to the experimental errors, yielding to fidelities of less than one [11].

In such cases where they have access only to a partially entangled state ρ, it is desirable to access a channel that allows a more faithful way to send quantum information. One solution is to implement protocols to increase the amount of entanglement [12,13]. These protocols are known as entanglement purification or entanglement distillation [14,15,16] and entanglement concentration [17]. These methods are based on the fact that local operations and classical communication between Alice and Bob cannot increase, on average, the amount of entanglement in the initially entangled pairs [18].

In the case of entanglement purification, the goal is to increase the purity and the entanglement in the initial state ρ, but under the cost of reducing the number of initial copies available. It can only be implemented successfully in a probabilistic way [14]. Moreover, an experimental realization of entanglement purification was carried out for mixed states of polarization-entangled photons using linear optics [19].

In the entanglement concentration, the process considers the cases where the initial partially entangled state is pure [20,21]. Indeed, there are two ways to implement entanglement concentration: the Procrustean method and the Schmidt projection method [17,20,21]. The Procrustean method is easier to implement than the Schmidt projection method because the initial partially entangled state is known. The entanglement concentration procedure is carried out by local filtering onto individual pairs of the initial state [17]. In the Schmidt method, however, the process of entanglement concentration is implemented in at least two unknown partially entangled states through collective simultaneous measurements onto the particles [22]. Thus, schemes for carrying out the entanglement concentration have been proposed for the Procrustean [23] and the Schmidt method [24,25]. Moreover, its experimental implementation has been achieved in the case of the Procrustean method [26] and for the Schmidt method [22] using partially polarization-entangled photons.

The entanglement concentration can also be classified as deterministic [12,27,28] as well as probabilistic [11,13,20,29]. In the deterministic case, the process has a probability equal to one to be successfully implemented in the regimes of few copies or in the asymptotic limit of infinite copies [30]. In this scheme, the quantum circuits to carry out deterministic entanglement concentration have been proposed [31] and recent experimental efforts demonstrate its feasibility [32,33,34,35]. In these experimental works, the copies are replaced by additional degrees of freedom of the same pair of photons, which improves the possibility of short-term implementations of entanglement distillation for technological purposes. On the other hand, in the probabilistic entanglement concentration, the process is achieved with a probability of less than one and has been experimentally implemented [36]. Moreover, the relation in the asymptotic limit between the entanglement concentration in a deterministic and probabilistic way was studied [30]. They found that these methods are equivalent considering many copies of the initial state: the error probability for the probabilistic method goes to zero quickly with the number of copies. In addition, the entanglement concentration is generally studied considering two entangled quantum states, but has also been studied for the case of tripartite correlated systems [37,38].

In this work, we studied the probabilistic entanglement concentration in the bipartite scenario of a pure two-qudit (*D*-dimensional) state. Considering a large dimensionality (D≫2), we study two methods to achieve entanglement concentration regarding a reasonably good probability of success at the expense of having a non-maximal entanglement. At first glance, we consider a tradeoff between the amount of entanglement of the state after the concentration procedure and its success probability, quantified by the payoff function Q. This figure of merit leads to analytically solving a quadratic optimization problem, ensuring that an optimal scheme for entanglement concentration can always be found in terms of Q. Then, a second method was studied, where we fixed the success probability and searched for the maximum amount of entanglement attainable in this case. We found that both ways resemble the Procrustean method applied to a subset of the most significant Schmidt coefficients without the constraint of obtaining a MES. We envisage the usefulness of these methods in entanglement-based quantum communication and also for device-independent protocols where high-dimensional entangled states are required with a certain amount of entanglement, such as randomness certification, expansion and self-testing [39,40,41].

## 2. Revisiting Entanglement Concentration

Throughout this work, we will limit ourselves to the case of entanglement concentration from a single copy of a two-qudit non-maximally entangled pure state. This state will be given by
(1)$$\begin{array}{c} {|\Phi \rangle }_{12}=\sum_{m=1}^{D}a_{m}{|m\rangle }_{1}{|m\rangle }_{2}, \end{array}$$
where $a_{m}$ are positive coefficients such that $\sum_{m}a_{m}^{2}=1$. The set of states ${\left\{{|m\rangle }_{1}{|m\rangle }_{2}\right\}}_{m=1}^{D}$ can be regarded as the Schmidt basis for the entangled state ${|\Phi \rangle }_{12}$, and, therefore, $a_{m}$ will be the respective Schmidt coefficients. In order to quantify the entanglement conveyed by ${|\Phi \rangle }_{12}$, the I-Concurrence [42] can be used, which is given by
(2)$$\begin{array}{cc} C({|\Phi \rangle }_{12}) & =\;\sqrt{\frac{D}{D-1}\left(1-\mathrm{tr}\left(\rho_{1}^{2}\right)\right)}\\ & =\;\sqrt{\frac{D}{D-1}\left(1-\sum_{m=1}^{D}a_{m}^{4}\right)}, \end{array}$$
where $\rho_{1}$ is the reduced density matrix of one of the qudits. This function fulfills the necessary conditions an entanglement measure needs to satisfy [43]. Its minimum value is zero, and its maximum is one, which arises when ${|\Phi \rangle }_{12}$ is a product state and a maximally entangled state, respectively. This document will refer to C simply as entanglement. Another function widely used to assess entanglement is the Schmidt number [44,45,46,47,48,49,50,51], defined as
(3)$$\begin{array}{c} K\left({|\Phi \rangle }_{12}\right)=\frac{1}{\mathrm{tr}\left(\rho_{1}^{2}\right)}={\left[\sum_{m=1}^{D}a_{m}^{4}\right]}^{-1}. \end{array}$$ It is straightforward to see that $C({|\Phi \rangle }_{12})$ and $K({|\Phi \rangle }_{12})$ are closely related, as both depend on $\mathrm{tr}\left(\rho_{1}^{2}\right)$.

As we mentioned above, it is well known the correlated state given in Equation (Equation 1) can have its entanglement increased through an entanglement concentration procedure [7,13,30,52,53]. This process is, in general, a probabilistic one [54]. We will follow the next approach to show the concentration scheme. Assuming we have an ancillary qubit initially prepared in state ${|0\rangle }_{a}$, it can be used for concentration through a unitary bipartite operation $U_{a1}$ acting over the ancilla and one of the qudits. Let
(4)$$\begin{array}{c} U_{a1}⊗I_{2}{|0\rangle }_{a}{|\Phi \rangle }_{12}={|0\rangle }_{a}A_{s}{|\Phi \rangle }_{12}+{|1\rangle }_{a}A_{f}{|\Phi \rangle }_{12}, \end{array}$$
where ${|\mu \rangle }_{a}$ is the state of the ancilla which flags whether concentration was accomplished (μ=0) or not (μ=1). $A_{s}$ and $A_{f}$ are Kraus operators acting on qudit 1, modifying the entangled state in each case. A measurement on the ancilla announces if we succeeded. Through this work, we will be only concerned with the successful cases, whose study can be simplified considering $A_{s}{|\Phi \rangle }_{12}$ only. Without loss of generality, we may write
(5)$$\begin{array}{c} A_{s}{|\Phi \rangle }_{12}=\sqrt{p_{s}}{|\Psi \rangle }_{12}, \end{array}$$
where $p_{s}$ is the probability of success for the concentration procedure, and ${|\Psi \rangle }_{12}$ is the resulting state; therefore, we obtain $C\left({|\Psi \rangle }_{12}\right)>C\left({|\Phi \rangle }_{12}\right)$. If the intention is to obtain a MES, it is known that $p_{s}=Da_{\min}^{2}$, where $a_{\min}^{2}=\min\{|a_{m}{|}^{2}\}$ [7,53]. This probability, however, may adopt very small values if the Schmidt coefficients exhibit large differences among them, rendering the procedure inefficient.

Alternatively, one may increase the success probability at expense of having a partially entangled state as result. In Ref. [20], Vidal studied the case of transforming Schmidt coefficients $\left\{a_{m}\right\}$ onto a given set $\left\{b_{m}\right\}$ and showed the optimal probability of success for such map. In this way, one may choose the $b_{m}$ coefficients in such a way that the success probability is *good enough* at the same time the entanglement is increased. Another possibility is to set the resulting state ${|\Psi \rangle }_{12}$ as a maximally entangled one *for a subspace* of dimension N⩽D, which is analogous to a Procrustean method (i.e., cutting off extra probabilities from a given reference value [14]) applied only on a subset of the original Schmidt coefficients [13]. Both approaches, however, force one to constrain the final state to be a given one. Thus, the problem contains *D* arbitrary parameters $b_{m}$, and one has to search thoroughly for a convenient combination of the $b_{m}$.

A possible way to decrease the number of free parameters is to use the Kraus operator $A_{s}\left(\xi \right)$ given in Ref. [55]. This approach allows to interpolate between the initial Schmidt coefficients ($a_{m}$) and the ones from a maximally entangled state ($1/\sqrt{D}$) using a single parameter ξ. Thus, we may transform $a_{m}\to b_{m}\left(\xi \right)$, where 0⩽ξ⩽1, and
(6)$$\begin{array}{c} b_{m}^{2}\left(\xi \right)=a_{m}^{2}+\left(\frac{1}{D}-a_{m}^{2}\right)\xi . \end{array}$$ It can be seen that Equation (Equation 6) shows a transformation that preserves the norm of the new state and represents a linear interpolation for the squares of the Schmidt coefficients. Moreover, the success probability is $p\left(\xi \right)={\left[1-\xi +\xi /(Da_{\min}^{2})\right]}^{-1}$ [55]. This method, although straightforward to understand, leads to little improvement in terms of success probabilities. For instance, Figure 1 evidences that even a little improvement in any of the functions used to assess entanglement is achieved at the expense of a substantial drop in the success probability. This figure also evidences that the I-Concurrence, although simple to work with because it is not a rational function, is not good for graphical assessment since even initial I-Concurrence (see ξ=0) exhibits values close to one. Instead, the Schmidt number is not simple to work with due to its inverse dependence on $\mathrm{tr}\left(\rho_{1}^{2}\right)$ but makes graphical evaluation uncomplicated.

These previous attempts lead us to question whether a method can obtain a *reasonable* increment in entanglement with a non-negligible success probability without imposing constraints on the final state beforehand. The next sections will address this question.

## 3. Towards Efficient Entanglement Concentration

Here, we shall propose and analyze a more efficient method for entanglement concentration from a single copy of a partially entangled pure state. Let us define a parameterized Kraus operator $A_{s}\left(\vec{z}\right)$ being applied on one of the qudits. This operator can be written as
(7)$$\begin{array}{c} A_{s}\left(\vec{z}\right)=\sum_{m=1}^{D}z_{m}\left|m\rangle \langle m\right|, \end{array}$$
so its action on the two-qudit system after successful concentration will be
(8)$$\begin{array}{cc} A_{s}\left(\vec{z}\right){|\Phi \rangle }_{12}= & \;\sum_{m=1}^{D}z_{m}a_{m}{|m\rangle }_{1}{|m\rangle }_{2}. \end{array}$$ Thus, keeping Equation (Equation 5) in mind, the post-concentration state and its probability of success are
(9)$$\begin{array}{cc} {|\Psi (\vec{z})\rangle }_{12}= & \;\frac{1}{\sqrt{p_{s}\left(\vec{z}\right)}}\sum_{m=1}^{D}a_{m}z_{m}{|m\rangle }_{1}{|m\rangle }_{2}, \end{array}$$
(10)$$\begin{array}{cc} p_{s}\left(\vec{z}\right)= & \;\sum_{m=1}^{D}a_{m}^{2}{\left|z_{m}^{\;}\right|}^{2}, \end{array}$$
respectively. Since $p_{s}\left(\vec{z}\right)$ must not exceed one, it is mandatory to impose $\left|z_{m}\right|⩽1$. The reduced density matrix for one of the subsystems shall be
(11)$$\begin{array}{c} \rho_{1}^{\;}\left(\vec{z}\right)=\frac{1}{p_{s}\left(\vec{z}\right)}\sum_{m=1}^{D}a_{m}^{2}{\left|z_{m}\right|}^{2}\left|m\rangle \langle m\right|. \end{array}$$ I-Concurrence and Schmidt number, as function of $\vec{z}$, will be given by
(12)$$\begin{array}{cc} C(\vec{z})= & \;\sqrt{\frac{D}{D-1}\left(1-\sum_{m=1}^{D}\frac{a_{m}^{4}{\left|z_{m}^{\;}\right|}^{4}}{p_{s}^{2}\left(\vec{z}\right)}\right)}, \end{array}$$
(13)$$\begin{array}{cc} K(\vec{z})= & \;\frac{p_{s}^{2}\left(\vec{z}\right)}{\sum_{m}a_{m}^{4}{\left|z_{m}^{\;}\right|}^{4}}. \end{array}$$

Let us now define a quantity $Q(\vec{z})$ aimed to assess the efficiency of the concentration procedure considering a trade-off between the probability of success and the increment in entanglement. A Kraus operator that maximizes this efficiency will be pursued. A choice, although not unique at all, might be $p_{s}\left(\vec{z}\right)C\left(\vec{z}\right)$. Maximizing it will be equivalent to maximizing its square, ${\left[p_{s}\left(\vec{z}\right)C\left(\vec{z}\right)\right]}^{2}$, which should be a simpler procedure since the square root we can see in Equation (Equation 12) will not be present. However, ${\left[p_{s}\left(\vec{z}\right)C\left(\vec{z}\right)\right]}^{2}$ has its maximum when $z_{m}=1,\;\forall \;m$, which means state ${|\Phi \rangle }_{12}$ will be kept unaltered (This will be proven in Appendix A). Instead, we may try with the difference between $C^{2}\left(\vec{z}\right)$ and a constant reference level for the I-Concurrence ($C_{\mathrm{REF}}^{2}$). This reference level could be, for instance, the initial value $C_{\mathrm{INIT}}=C\left({|\Phi \rangle }_{12}\right)$. Let us try by defining an efficiency function such as
(14)$$\begin{array}{c} Q\left(\vec{z}\right)=p_{s}^{2}\left(\vec{z}\right)\left(C^{2}\left(\vec{z}\right)-C_{\mathrm{REF}}^{2}\right). \end{array}$$ Equations (Equation 10) and (Equation 12) allow us to transform Equation (Equation 14) into
(15)$$\begin{array}{cc} Q(\vec{z})= & \frac{D}{D-1}\sum_{m,n=1}^{D}{\left|z_{m}\right|}^{2}a_{m}^{2}\left(P_{\mathrm{REF}}-\delta_{mn}\right)a_{n}^{2}{\left|z_{n}\right|}^{2}, \end{array}$$
(16)$$\begin{array}{cc} P_{\mathrm{REF}}= & 1-\frac{D-1}{D}C_{\mathrm{REF}}^{2}, \end{array}$$
where $P_{\mathrm{REF}}$ has been defined for mathematical convenience. It ranges from 1/D to 1, and it can be interpreted as a reference value for the purity of a reduced density matrix, as it can be seen from Equation (Equation 2). Another interpretation, as can be seen from Equation (Equation 3), is $P_{\mathrm{REF}}=1/K_{\mathrm{REF}}$, where $K_{\mathrm{REF}}$ is a reference value for the Schmidt number. A careful observation of Equation (Equation 15) leads us to infer that the problem of efficient entanglement concentration, in the form it has been described in this document, can be rewritten as a quadratic optimization problem given by
(17a)$$\begin{array}{cc} \max_{\vec{y}}Q\left(\vec{y}\right)= & \frac{D}{D-1}{\vec{y}}_{}^{⊺}H\vec{y}, \end{array}$$
(17b)$$\begin{array}{cc} \mathrm{subject}\;\mathrm{to}\; & 0⩽y_{m}⩽1, \end{array}$$
where
(17c)$$\begin{array}{cc} y_{m}= & {\left|z_{m}\right|}^{2}, \end{array}$$
(17d)$$\begin{array}{cc} {\left[H\right]}_{m,n}= & \left(P_{\mathrm{REF}}-\delta_{mn}\right)a_{m}^{2}a_{n}^{2}. \end{array}$$

Therefore, the problem of efficient entanglement concentration for a single pair of entangled qudits can be written as the quadratic optimization problem described in Equations ([Disp-formula FD17a-micromachines-14-01207])–([Disp-formula FD17d-micromachines-14-01207]), with the optimization variables $y_{m}$ lying in a unit hypercube. Finally, without loss of generality, we may choose the positive root of $z_{m}=\sqrt{y_{m}}$. Note that the presence of $C_{\mathrm{REF}}$ forces the optimization to look for a solution ${\vec{y}}_{\mathrm{OPT}}$ such that $C\left({\vec{y}}_{\mathrm{OPT}}\right)⩾C_{\mathrm{REF}}$. Otherwise, function $Q({\vec{y}}_{\mathrm{OPT}})$ would adopt a negative value [see Equation (Equation 14)] and, therefore, it will not represent a maximum. For this reason, we can assure that $C_{\mathrm{REF}}⩾C_{\mathrm{INIT}}$ forces entanglement concentration. In an extreme case, $C_{\mathrm{REF}}=1$ means that the reference level is equal to the maximum possible value I-Concurrence can adopt. Therefore, $Q(\vec{y})$ will adopt a negative value *unless* the final entanglement is also equal to one, for which Q=0. This is the standard entanglement concentration procedure. On the other hand, $C_{\mathrm{REF}}$ could be slightly smaller than $C_{\mathrm{INIT}}$ and, still, entanglement concentration may occur, as it will be shown in Section 4.1. For this problem, the square of the I-Concurrence has been used also because both numerical and analytical solutions are accessible. For graphical purposes, as it was already seen in Figure 1, the Schmidt number shall be used. Moreover, the Schmidt number provides an estimation of the number of relevant Schmidt modes involved [45].

We must add that the Kraus operator defined in Equation (Equation 7) is diagonal in the Schmidt basis. We may have started by a general Kraus operator, instead of a diagonal one. However, Appendix B shows it suffices to look for diagonal operators.

## 4. Solving the Problem

### 4.1. Numerical Hints

Figure 2 shows the results of numerical resolution of the aforementioned optimization problem for a given set of D=16 Schmidt coefficients $a_{m}^{2}$, randomly chosen, and sorted decreasingly in order to ease observation. For this example, we tested four possible values of $C_{\mathrm{REF}}^{2}$, given by (i) $C_{\mathrm{INIT}}^{2}/2$, much smaller than the initial entanglement; (ii) $0.98C_{\mathrm{INIT}}^{2}$, slightly smaller than the initial entanglement; (iii) average value between $C_{\mathrm{INIT}}$ and 1, a significant increase in entanglement; and (iv) $C_{\mathrm{REF}}^{2}=1$, the maximum possible value for $C_{\mathrm{REF}}^{2}$. The optimization was performed using the function quadprog of Matlab R2022b. Since this is a non-convex problem with constant bounds only, the algorithm “trust-region-reflective” was used since it was the best suited for our optimization problem [56].

The plots show the original Schmidt coefficients (cyan) and the non-normalized coefficients after concentration (dark red). A pattern is evident. For small values of $C_{\mathrm{REF}}^{2}$, keeping the state as it is seems to be the best option in terms of efficiency. As $C_{\mathrm{REF}}^{2}$ increases, the solutions of the optimization problem suggest one to use a Procrustean method on the *n* largest Schmidt coefficients, where *n* increases according $C_{\mathrm{REF}}^{2}$ moves closer to one. This is analogous to entanglement concentration on a subspace of the bipartite Hilbert space as the one proposed in Ref. [13], although we have not required the final state to be fixed to a given one. Finally, $C_{\mathrm{REF}}^{2}=1$ represents the ideal entanglement concentration context, in which the resulting state exhibits the maximal entanglement possible. The optimization problem shows the correct result, which consists of uniforming all post-concentration Schmidt coefficients.

Although Figure 2 shows a single set of initial Schmidt coefficients, the same pattern is observed for other states in any dimension D>2. In the following, we shall prove why the Procrustean method on a subspace is the most efficient method, according to our figures of merit.

### 4.2. Analytical Results

One of the goals of this work is to find the analytical solution of the optimization problem of Equation (17). The details of the proof will be shown in the next subsections. The procedure can be summarized as follows:If $P_{\mathrm{REF}}=1/D$ (minimum attainable value, equivalent to $C_{\mathrm{REF}}=1$), it means we are pursuing a standard entanglement concentration using all Schmidt coefficients. Then, perform concentration using $z_{m}=a_{\min}/a_{m}$. Otherwise, follow Steps 2–8.Sort the Schmidt coefficients in decreasing order. Let us label these sorted coefficients as $a_{m}$.Define a vector $\vec{\beta }$ such that $\beta_{n}=1-\sum_{m=1}^{n}a_{m}^{2}$, for n=1,…,D.Define a vector $\vec{\alpha }$ such that $\alpha_{n}=P_{\mathrm{REF}}\beta_{n}/\left(1-nP_{\mathrm{REF}}\right)$.Find the largest value of *n* that allow both $\alpha_{n}⩽a_{n}^{2}$ and $n<1/P_{\mathrm{REF}}$ to be simultaneously satisfied. Let us label this value as $n_{\mathrm{OPT}}$.Define $\vec{x}$ such that
$$\begin{array}{c} x_{m}=\left\{\begin{array}{cc} \alpha_{n_{\mathrm{OPT}}}^{\;} & ,\;\mathrm{for}\;m=1,\ldots ,n_{\mathrm{OPT}},\\ 1 & ,\;\mathrm{for}\;m=n_{\mathrm{OPT}}+1,\ldots ,D. \end{array}\right. \end{array}$$Define $y_{m}=x_{m}/a_{m}^{2}$. Afterwards, sort the $y_{m}$ using the inverse of the sorting operation described in Step 1. These sorted values will be the $y_{m}$ that solve the optimization problem of Equation (17).Define $z_{m}=\sqrt{y_{m}}$. These values are the ones needed to construct the Kraus operator of Equation (Equation 7).

Section 4.2.1, Section 4.2.2, Section 4.2.3, Section 4.2.4, Section 4.2.5, Section 4.2.6 and Section 4.2.7 hereunder shall detail the underlying reasoning for the algorithm shown above.

#### 4.2.1. Redefining the Optimization Problem

In order to prove the solution detailed above, we shall define $x_{m}=a_{m}^{2}y_{m}^{\;}=a_{m}^{2}{\left|z_{m}\right|}^{2}$. This allows us to write the optimization problem [Equation (17)], up to a proportionality constant, in a simpler way: (18)$$\begin{array}{c} \left\{\begin{array}{cc} \max_{\vec{x}} & Q\left(\vec{x}\right)=P_{\mathrm{REF}}{\left(\sum_{m=1}^{D}x_{m}\right)}^{2}-\sum_{m=1}^{D}x_{m}^{2},\\ \;\; & \;\;\\ s.\;t. & 0⩽x_{m}⩽a_{m}^{2}. \end{array}\right. \end{array}$$ These new variables $x_{m}$ are the ones plotted in Figure 2 using dark red bars. Thus, the $x_{m}$ will provide an idea about the post-concentration Schmidt coefficients.

The domain is no longer the unit hypercube, but an orthotope whose vertices have coordinates components equal to zero and $a_{m}^{2}$. Thus, every $x_{m}$ has three options: (i) having a fixed value equal to zero, (ii) having a fixed value equal to $a_{m}^{2}$, and (iii) having a variable value between zero and $a_{m}^{2}$. These options had to be taken into account in order to find all critical points.

#### 4.2.2. Finding Critical Points

For starters, we shall define set of indices according to the aforementioned options:$Z=\{j:x_{j}=0\}$;$O=\{k:x_{k}=a_{k}^{2}\}$;$I=\{\ell :0<x_{\ell }<a_{\ell }^{2}\}$.

The symbols Z, O, and I stand for *zero*, *outer*, and *inner*, respectively. In this way, any summation can be written as $\sum_{m}=\sum_{j\in Z}+\sum_{k\in O}+\sum_{\ell \in I}$. There exist $3^{D}$ configurations for (Z,O,I). If we label each of those $3^{D}$ combinations by using the index μ, then we can define function $Q_{\mu }\left(\vec{x}\right)$ as the function $Q(\vec{x})$ for the μth configuration. Explicitly,
(19)$$\begin{array}{cc} Q_{\mu }\left(\vec{x}\right)= & \;P_{r}ef^{\;}{\left(\sum_{k\in O_{\mu }}a_{k}^{2}+\sum_{\ell \in I_{\mu }}x_{\ell }\right)}^{2}-\sum_{k\in O_{\mu }}a_{k}^{4}-\sum_{\ell \in I_{\mu }}x_{\ell }^{2}. \end{array}$$By imposing $\partial_{x_{r}}Q_{\mu }\left(\vec{x}\right)=0$, we can find the critical points of $Q_{\mu }\left(\vec{x}\right)$. Consequently,
(20)$$\begin{array}{c} x_{r}=P_{\mathrm{REF}}\left(\sum_{k\in O_{\mu }}a_{k}^{2}+\sum_{\ell \in I_{\mu }}x_{\ell }\right),\;\;\;r\in I_{\mu }. \end{array}$$This means that as long as $x_{r}$ is not fixed in either 0 or $a_{r}^{2}$, the optimal solution is such that those $x_{r}$ all adopt the same value. Let us define some additional ancillary parameters,
(21)$$\begin{array}{c} \beta_{\mu }=\sum_{k\in O_{\mu }}a_{k}^{2},\;\;\;\;\;\;\;\gamma_{\mu }=\sum_{k\in O_{\mu }}a_{k}^{4},\;\;\;\;\;\;\;n_{\mu }=\left|I_{\mu }\right|, \end{array}$$
$n_{\mu }$ being the number of free parameters $x_{\ell }$. With these definitions, we can now assert that $x_{\ell }=\alpha_{\mu }$ is the critical point for the μth configuration, where
(22)$$\begin{array}{c} x_{\ell }=\alpha_{\mu }=\frac{P_{\mathrm{REF}}\beta_{\mu }}{1-P_{\mathrm{REF}}n_{\mu }},\;\;\;\;\;\;\forall \;\ell \in I_{\mu }. \end{array}$$ Consequently, if $Q_{\mu }$ is the value of $Q_{\mu }\left(\vec{x}\right)$ evaluated at the μth critical point, then
(23)$$\begin{array}{cc} Q_{\mu }= & P_{\mathrm{REF}}{\left(\beta_{\mu }+n_{\mu }\alpha_{\mu }\right)}^{2}-\gamma_{\mu }-n_{\mu }\alpha_{\mu }^{2}\\ = & \;\alpha_{\mu }\beta_{\mu }-\gamma_{\mu }. \end{array}$$ The fact that $x_{\ell }=\alpha_{\mu }$ means that, for every $\ell \in I_{\mu }$, coefficients $a_{\ell }^{2}$ will be transformed into $\alpha_{\mu }$ as consequence of the concentration procedure. This is, precisely, the Procrustean method applied on a $n_{\mu }$-dimensional subset of the coefficients $\left\{a_{m}\right\}$.

It is worth mentioning that Equation (Equation 22) contains the implicit assumption $P_{\mathrm{REF}}\neq 1/n_{\mu }$, which raises questions regarding the case $P_{\mathrm{REF}}=1/n_{\mu }$. If that were the case, trying to solve Equation (Equation 20) leads us to conclude $\beta_{\mu }=0$ and, equivalently, $O_{\mu }=\emptyset$. In turn, this implies $Q_{\mu }\left(\vec{x}\right)=0$. Nevertheless, we may see from the original definition of $Q(\vec{z})$ [Equation (Equation 14)] that the only possible way in which $Q_{\mu }\left(\vec{x}\right)=0$ represents a maximum occurs when $C_{\mathrm{REF}}^{2}=1$ and $C^{2}\left(\vec{z}\right)=1$ simultaneously, i.e., $P_{\mathrm{REF}}=1/D$ has been set and the resulting state is a *D*-dimensional maximally entangled state.

#### 4.2.3. Upper Bounds for $n_{\mu }$

The Hessian matrix has components given by
(24)$$\begin{array}{c} \partial_{x_{s}}\partial_{x_{r}}Q_{\mu }\left(\vec{x}\right)=2\left(P_{\mathrm{REF}}-\delta_{rs}\right). \end{array}$$ It can be shown that $Q_{\mu }$ will represent a local maximum for the μth configuration provided, $(1-n_{\mu }P_{\mathrm{REF}})>0$, since this condition ensures Hessian matrix to be negative-definite. In other words,
(25)$$\begin{array}{c} n_{\mu }<\frac{1}{P_{\mathrm{REF}}}. \end{array}$$ Thus, some configurations $\left(Z_{\mu },O_{\mu },I_{\mu }\right)$ can be immediately discarded if $n_{\mu }$ exceeds this bound.

#### 4.2.4. Eliminating Zeros

Let us start by analyzing the effect of zeros by comparing a given $Q_{\mu }$—for which $x_{r}=0$—with the value of $Q_{\mu^{\prime }}\left(\vec{x}\right)$ when $x_{r}=\delta ⪆0$. Using Equation (Equation 19), we have that
(26)$$\begin{array}{cc} Q_{\mu }{|}_{x_{r}=0}= & P_{\mathrm{REF}}{\left(\beta_{\mu }+n_{\mu }\alpha_{\mu }\right)}^{2}-\gamma_{\mu }-n_{\mu }\alpha_{\mu }^{2}, \end{array}$$
(27)$$\begin{array}{cc} Q_{\mu^{\prime }}{|}_{x_{r}\to \delta }= & P_{\mathrm{REF}}{\left(\beta_{\mu }+n_{\mu }\alpha_{\mu }+\delta \right)}^{2}-\gamma_{\mu }-n_{\mu }\alpha_{\mu }^{2}-\delta^{2}, \end{array}$$
which, in turn, leads us to
(28)$$\begin{array}{cc} Q_{\mu^{\prime }}{|}_{x_{r}\to \delta }-Q_{\mu }{|}_{x_{r}=0}= & 2P_{\mathrm{REF}}\left(\beta_{\mu }+n_{\mu }\alpha_{\mu }\right)\delta +O\left(\delta^{2}\right)>0. \end{array}$$ We can see that $Q_{\mu^{\prime }}$ actually grows if $x_{r}$ moves away from zero within its neighborhood. This means that every configuration containing a null value on *any* of its $x_{m}$ cannot represent a maximum since all neighboring points have higher values for $Q(\vec{x})$. Therefore, the solution we are looking for is such that $Z_{\mu }=\emptyset$. The number of remaining configurations is now less than $2^{D}$.

#### 4.2.5. Optimal *n* Will Be the Largest Possible

We are left with the options $x_{m}\in \left\{\alpha_{\mu },a_{m}^{2}\right\}$. We know that the μth critical point is such that $x_{\ell }=\alpha_{\mu },\;\;\forall \;\ell \in I_{\mu }$. Since $\vec{x}$ still belongs to the orthotope, an additional condition arises: $\alpha_{\mu }⩽a_{\ell }^{2},\;\;\forall \;\ell \in I_{\mu }$.

Let us now compare two solutions $Q_{\lambda }$ and $Q_{\nu }$, whose critical points differ only in one term $x_{r}$, so $r\in O_{\lambda }$ and $r\in I_{\nu }$. Thus, by using Equations (Equation 21)–(Equation 23), we have that
(29)$$\begin{array}{cc} \beta_{\nu }= & \beta_{\lambda }-a_{r}^{2}, \end{array}$$
(30)$$\begin{array}{cc} \gamma_{\nu }= & \gamma_{\lambda }-a_{r}^{4}, \end{array}$$
(31)$$\begin{array}{cc} n_{\nu }= & n_{\lambda }+1, \end{array}$$
(32)$$\begin{array}{cc} \alpha_{\nu }= & \frac{P_{\mathrm{REF}}\left(\beta_{\lambda }-a_{r}^{2}\right)}{1-\left(n_{\lambda }+1\right)P_{\mathrm{REF}}}, \end{array}$$
(33)$$\begin{array}{cc} Q_{\lambda }= & \alpha_{\lambda }\beta_{\lambda }-\gamma_{\lambda }, \end{array}$$
(34)$$\begin{array}{cc} Q_{\nu }= & \alpha_{\nu }\beta_{\nu }-\gamma_{\nu }. \end{array}$$ Consequently,
(35)$$\begin{array}{cc} Q_{\lambda }-Q_{\nu }= & -\frac{{\left(P_{r}ef^{\;}\beta_{\lambda }-\left(1-n_{\lambda }P_{r}ef^{\;}\right)a_{r}^{2}\right)}^{2}}{\left(1-n_{\lambda }P_{r}ef^{\;}\right)\left(1-\left(n_{\lambda }+1\right)P_{r}ef^{\;}\right)}<0. \end{array}$$ Therefore, a better solution is obtained when *r* belongs to $I_{\nu }$ over $O_{\lambda }$, provided that the constraints are fulfilled. In simpler words, the best of the $\left\{n_{\mu }\right\}$ will be the largest possible within the conditions $n_{\mu }<1/P_{\mathrm{REF}}$ and $\alpha_{\mu }⩽a_{\ell }^{2},\;\;\forall \;\ell \in I_{\mu }$.

#### 4.2.6. Sorting Preference

For the following comparison, it will be helpful to define two sets $O_{0}$ and $I_{0}$. We will center our attention on two values $x_{r}$ and $x_{s}$. Now, let us compare two solutions $Q_{\rho }$ and $Q_{\sigma }$ that satisfy
(36)$$\begin{array}{cc} n_{\rho }=n_{\sigma }=n, & \end{array}$$
(37)$$\begin{array}{cc} I_{\rho }=I_{0}\cup \left\{r\right\},\;\;\; & I_{\sigma }=I_{0}\cup \left\{s\right\}, \end{array}$$
(38)$$\begin{array}{cc} O_{\rho }=O_{0}\cup \left\{s\right\},\;\;\; & O_{\sigma }=O_{0}\cup \left\{r\right\}. \end{array}$$ Thus, $I_{\rho }$ and $I_{\sigma }$ have n−1 elements in common, whereas $O_{\rho }$ and $O_{\sigma }$ have D−n−1 elements in common. Consequently,
(39)$$\begin{array}{cc} \beta_{\rho }=\beta_{0}+a_{s}^{2},\;\;\; & \gamma_{\rho }=\gamma_{0}+a_{s}^{4}, \end{array}$$
(40)$$\begin{array}{cc} \beta_{\sigma }=\beta_{0}+a_{r}^{2},\;\;\; & \gamma_{\sigma }=\gamma_{0}+a_{r}^{4}, \end{array}$$
where $\beta_{0}=\sum_{k\in O_{0}}a_{k}^{2}$ and $\gamma_{0}=\sum_{k\in O_{0}}a_{k}^{4}$. For the following, we shall assume $a_{r}>a_{s}$. Now, since both $Q_{\rho }$ and $Q_{\sigma }$ are admissible solutions, it *must* happen that $\alpha_{\rho }⩽a_{r}^{2}$ and $\alpha_{\sigma }⩽a_{s}^{2}$ as consequence of Equations (Equation 18), (Equation 22), and (Equation 37). This means
(41)$$\begin{array}{c} t\left(\beta_{0}+a_{s}^{2}\right)⩽a_{r}^{2},\;\;\;\;\;\mathrm{and}\;\;\;\;\;t\left(\beta_{0}+a_{r}^{2}\right)⩽a_{s}^{2}, \end{array}$$
where $t=P_{\mathrm{REF}}/\left(1-nP_{r}ef^{\;}\right)$ is a positive parameter. If we add these two inequalities, we obtain
(42)$$\begin{array}{c} \left(a_{r}^{2}+a_{s}^{2}\right)\left(1-t\right)-2t\beta_{0}⩾0. \end{array}$$ The difference between the solutions $Q_{\rho }$ and $Q_{\sigma }$ is
(43)$$\begin{array}{cc} \Delta Q= & \;Q_{\rho }-Q_{\sigma }\\ = & \;\left(a_{r}^{2}-a_{s}^{2}\right)\left(\left(1-t\right)\left(a_{r}^{2}+a_{s}^{2}\right)-2t\beta_{0}\right). \end{array}$$ Since $a_{r}>a_{s}$ was assumed and the inequality of Equation (Equation 42) was obtained, it can be assured that $Q_{\rho }>Q_{\sigma }$. Now, let us remember that $Q_{\rho }$ is the solution in which $x_{r}=\alpha_{\rho }$ and $x_{s}=a_{s}^{2}$. This means it is better to cut off coefficient $a_{r}$ (the larger one) over $a_{s}$.

Since we already know (see Section 4.2.5) that *n* must be the largest possible within the constraints $n<1/P_{\mathrm{REF}}$ and $\alpha_{\mu }⩽a_{\ell }^{2},\;\;\forall \;\ell \in I_{\mu }$, we must compare now all the solutions $Q_{\mu }$ such that $n_{\mu }$ is equal to that optimal value of *n*. According to the computations of this section, the most efficient concentration scheme will consist in cutting off the *n* largest Schmidt coefficients, which is in complete agreement with the results shown in Figure 2.

#### 4.2.7. How to Construct the Optimal Concentration Scheme

In summary, we know now that if $C_{\mathrm{REF}}=1$ (equivalently, $P_{\mathrm{REF}}=1/D$), then the optimal solution corresponds to a entanglement concentration procedure that yields a *D*-dimensional maximally entangled state. On the other hand, if $C_{\mathrm{REF}}<1$ (equivalently, $P_{\mathrm{REF}}>1/D$), we have shown that the optimal solution (i) does not contain zeros, (ii) it has values either given by $x_{m}=a_{m}^{2}$ (i.e., keep $a_{m}$ as they are) or by $x_{m}=\alpha_{\mu }$ (i.e., crop coefficients $a_{m}$ to a given value $\alpha_{\mu }$), (iii) the *n* largest Schmidt coefficients are to be cropped, and (iv) *n* must be as large as possible within constraints given by $n<1/P_{\mathrm{REF}}$ and $\alpha_{\mu }⩽a_{m}^{2}$. Once the optimal $x_{m}$ are found, we may compute the corresponding $y_{m}$ and $z_{m}$. These rules gave rise to the algorithm described at the beginning of Section 4.2. Moreover, we performed thousands of numerical simulations, ranging from D=32 to D=1024, that confirmed such an algorithm actually provides the optimal solution. Figure 3 shows a sample of those simulations for D=1024, depicting relative differences between the results from numerical optimization (${\vec{y}}_{\mathrm{num}}$ and $Q({\vec{y}}_{\mathrm{num}})$) and the ones from the algorithm proposed in this section (${\vec{y}}_{\mathrm{alg}}$ and $Q({\vec{y}}_{\mathrm{alg}})$) for 100 values of $P_{\mathrm{REF}}$. These relative differences are computed as
(44)$$\begin{array}{cc} \Delta y_{\mathrm{relative}}= & \;\frac{1}{D}\sum_{m=1}^{D}\left|\frac{{\left({\vec{y}}_{\mathrm{num}}\right)}_{m}-{\left({\vec{y}}_{\mathrm{alg}}\right)}_{m}}{{\left({\vec{y}}_{\mathrm{num}}^{\;}\right)}_{m}}\right|, \end{array}$$
(45)$$\begin{array}{cc} \Delta Q_{\mathrm{relative}}= & \;\left|\frac{Q\left({\vec{y}}_{\mathrm{num}}\right)-Q\left({\vec{y}}_{\mathrm{alg}}\right)}{Q({\vec{y}}_{\mathrm{num}})}\right|. \end{array}$$ The initial Schmidt coefficients were computed from a randomly-generated D×D entangled state. As the data of Figure 3 show, relative differences between the two solutions being compared are negligible, thus demonstrating the adequateness of the proposed algorithm. Discrepancies can be explained as a consequence of floating-point computation precision.

After efficiency optimization, one should evaluate whether practical advantages were obtained from it. Figure 4 shows the probability of success and Schmidt number for the same optimizations carried out for Figure 3. The initial state had a Schmidt number $K_{\mathrm{INIT}}\approx 512$. Raising this number to its maximum (i.e., K=1024) can be done with a probability of success $p_{s}=Da_{\min}^{2}∼10^{-7}$ (not shown in the graphs in order to ease observation). However, non-maximal Schmidt numbers can be obtained with much better probabilities. For instance, $P_{\mathrm{REF}}\approx 1.15\times 10^{-3}$ allows one to achieve a considerable Schmidt number (K=900) with a success probability $p_{s}=11\%$. Although $P_{\mathrm{REF}}\approx 1.15\times 10^{-3}$ seems to be a non-trivial number of uncertain origin, we may notice that $1/P_{\mathrm{REF}}∼868$. Thus, an acceptable method to estimate the necessary value of $P_{\mathrm{REF}}$ consists in setting a minimum desirable Schmidt number $K_{\mathrm{MIN}}$, define a slightly smaller threshold number $K_{\mathrm{THR}}<K_{\mathrm{MIN}}$, and computing $P_{\mathrm{REF}}=1/K_{\mathrm{THR}}$.

It is worth mentioning that the solution described in this section closely resembles the entanglement concentration procedure described in Ref. [13], which was also graphically explained in Ref. [30]. However, we did not set the final state to a fixed one in our formulation. Instead, we defined a single figure of merit to be interpreted as efficiency, and its optimization suggested performing entanglement concentration on the subspace of the largest Schmidt coefficients.

## 5. Entanglement Concentration with Fixed Probability of Success

An alternative way to solve the problem of efficient entanglement concentration is by setting the success probability to a fixed value $p_{\mathrm{FIX}}$ and inquiring about the largest entanglement that can be extracted. As it can be seen from Equations (Equation 12) and (Equation 13), this question reduces to minimization of the purity of the reduced density matrix, as
(46)$$\begin{array}{cc} \min_{\vec{y}}\;P\left(\vec{y}\right)= & \;\left[\frac{1}{p_{s}\left(\vec{y}\right)}\sum_{m=1}^{D}a_{m}^{4}y_{m}^{2}\right],\\ \mathrm{subject}\;\mathrm{to}\; & \;0⩽y_{m}⩽1\;\;\;\mathrm{and}\;\;\;\sum_{m=1}^{D}a_{m}^{2}y_{m}=p_{\mathrm{FIX}}, \end{array}$$
where we have already used $y_{m}={\left|z_{m}\right|}^{2}$. As we have imposed $p_{s}\left(\vec{y}\right)=p_{\mathrm{FIX}}$, the optimization reduces to optimize $\sum_{m}a_{m}^{4}y_{m}^{2}$. As in the previous section, we shall resort to $x_{m}=y_{m}^{2}$ and the sets of indices $Z_{\mu }$, $O_{\mu }$, and $I_{\mu }$. Using the $x_{m}$, we are left to optimize $\sum_{m}x_{m}^{2}$, and the constraint of fixed probability can be rewritten as $\sum_{m}x_{m}=p_{\mathrm{FIX}}$, which also allows us to write one of the variables in terms of the others. Let
(47)$$\begin{array}{c} x_{\vartheta }=p_{\mathrm{FIX}}-\sum_{m\neq \vartheta }x_{m}. \end{array}$$ Then, the minimization of the purity can be rewritten as
(48)$$\begin{array}{cc} \mathrm{minimize}\left(p_{\mathrm{FIX}}P\left(\vec{x}\right)\right) & =\;\sum_{m\neq \vartheta }x_{m}^{2}+{\left(p_{\mathrm{FIX}}-\sum_{m\neq \vartheta }x_{m}\right)}^{2}\\ & =\;\sum_{k\in O_{\mu }}a_{k}^{4}+\sum_{\begin{array}{c} \ell \in I_{\mu }\\ \ell \neq \vartheta \end{array}}x_{\ell }^{2}+{\left(\sum_{k\in O_{\mu }}a_{k}^{2}+\sum_{\begin{array}{c} \ell \in I_{\mu }\\ \ell \neq \vartheta \end{array}}x_{\ell }\right)}^{2}. \end{array}$$

Critical points are found by setting $\partial \left(p_{\mathrm{FIX}}P\left(\vec{x}\right)\right)/\partial x_{r}=0$, with $r\in I_{\mu }$ and r≠ϑ. This leads us to $x_{r}=\kappa_{\mu }$, where
(49)$$\begin{array}{c} \kappa_{\mu }=\frac{p_{\mathrm{FIX}}-\beta_{\mu }}{n_{\mu }}. \end{array}$$ In turn, Equation (Equation 47) implies that $x_{\vartheta }=\kappa_{\mu }$ as well. Thus, we obtained solutions given by either $x_{m}=a_{m}^{2}$, $x_{m}=0$, or $x_{m}=\kappa_{\mu }$, which is the exact behavior exhibited by the $x_{m}$ from Section 4 up to a change from $\alpha_{\mu }$ to $\kappa_{\mu }$. The same analysis performed in Section 4.2.4, Section 4.2.5, Section 4.2.6 and Section 4.2.7 can be applied here. The conclusions are very similar: (i) the optimal values of $x_{m}$ are different from zero, (ii) if *n* is the number of variables $x_{m}$ being equal to $\kappa_{\mu }$, then *n* must be as large as possible within the constraint $0⩽\kappa ⩽a_{\ell }^{2}$, and (iii) the *n* largest Schmidt coefficients are cut off. Thus, an algorithm can be constructed as follows:Sort the Schmidt coefficients in decreasing order. Let us label these sorted coefficients as $a_{m}$.Define a vector $\vec{\beta }$ such that $\beta_{n}=1-\sum_{m=1}^{n}a_{m}^{2}$, for n=1,…,D.Define a vector $\vec{\kappa }$ such that $\kappa_{n}=\left(p_{\mathrm{FIX}}-\beta_{n}\right)/n$.Find the largest value of *n* such that $\kappa_{n}⩾0$ and $\kappa_{n}<a_{n}^{2}$ are simultaneously satisfied. Let us label this value as $n_{\mathrm{OPT}}$.Define $\vec{x}$ such that
$$\begin{array}{c} x_{m}=\left\{\begin{array}{cc} \kappa_{n_{\mathrm{OPT}}}^{\;} & ,\;\mathrm{for}\;m=1,\ldots ,n_{\mathrm{OPT}},\\ 1 & ,\;\mathrm{for}\;m=n_{\mathrm{OPT}}+1,\ldots ,D. \end{array}\right. \end{array}$$Define $y_{m}=x_{m}/a_{m}^{2}$. Afterwards, sort the $y_{m}$ using the inverse of the sorting operation described in Step 1. These sorted values will be the $y_{m}$ that solve the optimization problem of Equation (17).Define $z_{m}=\sqrt{y_{m}}$. These values are the ones needed to construct the Kraus operator of Equation (Equation 7).

As it can be seen, the solutions obtained for this problem are completely analogous to the ones of the previous section. The advantage of this approach lies in the fact that $P(\vec{x})$ appears in both I-Concurrence and Schmidt number. Thus, it is a favorable way to increase the Schmidt number without introducing nontrivial mathematical complications. Once more, this result represents a Procrustean method applied on a subspace, although only one parameter has been fixed ($p_{\mathrm{FIX}}$) instead of a whole state.

## 6. Conclusions

In summary, we have studied entanglement concentration from a single copy of a two-qudit entangled state in terms of efficiency. As the ideal procedure—obtaining a maximally entangled state—is extremely inefficient in terms of probability, we studied the possibility of concentrating a fair enough amount of entanglement and, simultaneously, increment the success probability. Two methods were analyzed. For the first one, a function $Q(\vec{y})$ was defined in order to quantify efficiency as the product of success probability and entanglement increment. This function allows one to introduce a parameter $P_{\mathrm{REF}}$, which is loosely related to a minimal entanglement amount intended to extract. The other one consisted of fixing the success probability to a given value and finding the maximal entanglement it can be extracted under the constraint herein. We found that, for both cases, the solution resembles a Procrustean method applied on a subset of the largest Schmidt coefficients. Such application of the Procrustean method has been already studied in the literature under the assumption that the final state *must* be a *n*-dimensional maximally entangled state, with n<D. Therefore, *n* constraints are implicitly assumed. Instead, this work does not impose constraints on the final state. In the first method, the Procrustean method results as consequence of a quadratic optimization problem. In the second one, it emerges after optimizing entanglement and using a single constraint.

We anticipate that this work may be useful for understanding how to concentrate entanglement efficiently in very large dimensions. As entanglement is a resource underlying many protocols in Quantum Information Science, we believe many people in the Quantum Information community may benefit from these findings.

## Figures and Tables

**Figure 1 micromachines-14-01207-f001:**
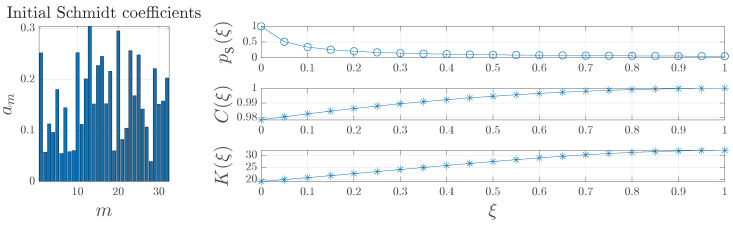
Example of entanglement concentration for D=32 by using linear interpolation for the squares of the Schmidt coefficients.

**Figure 2 micromachines-14-01207-f002:**
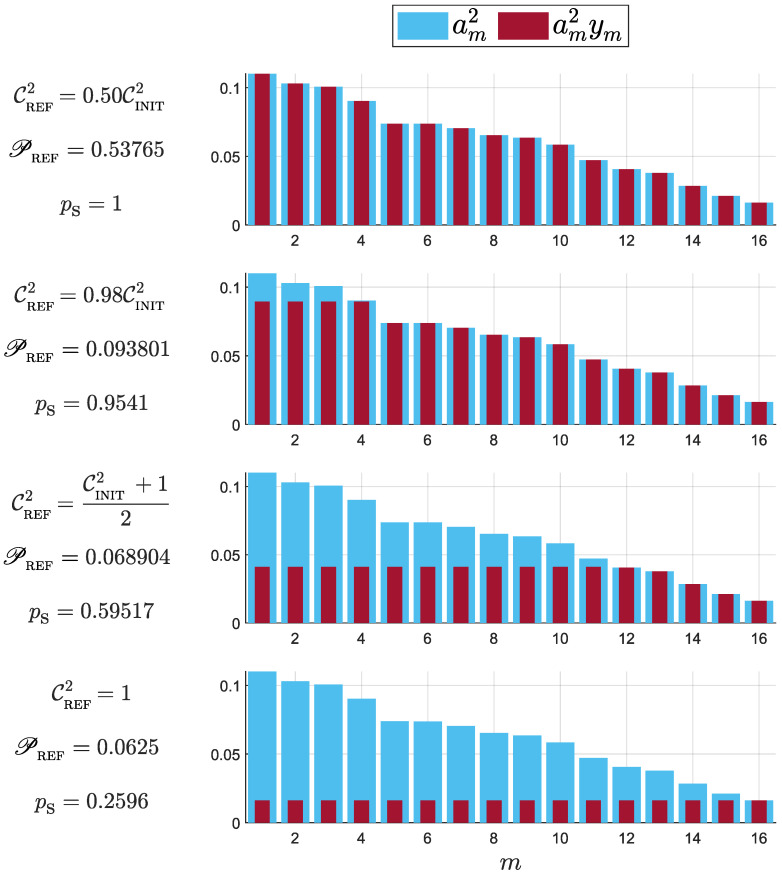
Numerical example of resolution of the quadratic optimization problem [Equation (17)] for dimension D=16, using 4 different values of $C_{\mathrm{REF}}^{2}$. Bars show the original Schmidt coefficients (cyan) and the non-normalized coefficients after concentration (dark red). Their respective values of $P_{\mathrm{REF}}$ and probabilities of success $p_{s}$ are also shown.

**Figure 3 micromachines-14-01207-f003:**
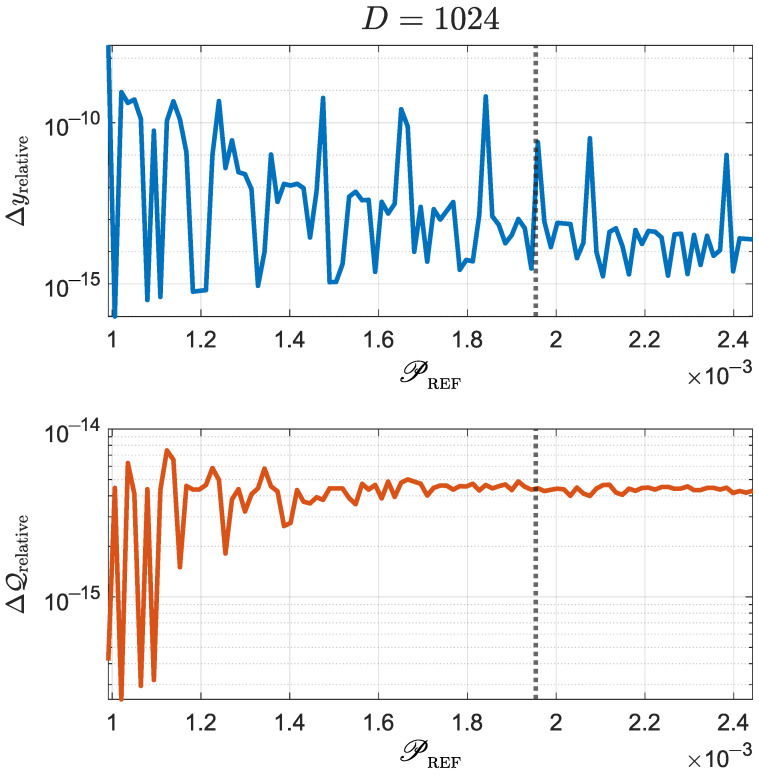
Comparison between results obtained through numerical optimization ($Q({\vec{y}}_{\mathrm{num}})$) and the ones obtained by using the algorithm introduced at the beginning of Section 4.2 ($Q({\vec{y}}_{\mathrm{alg}})$). Relative differences for are shown for 100 values of $P_{\mathrm{REF}}$. The vertical dotted line indicates the initial value of the purity of the reduced density matrix, i.e., $P_{\mathrm{REF}}=P_{\mathrm{INIT}}$. See the main text for details about the computation of these relative differences.

**Figure 4 micromachines-14-01207-f004:**
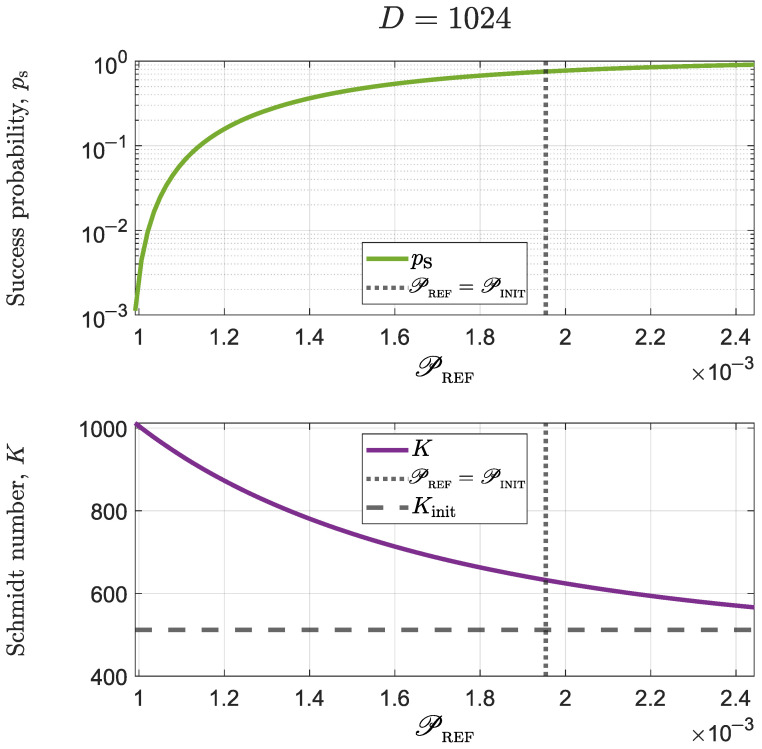
Success probability and Schmidt number for the same state and optimizations used in Figure 3. The vertical dotted line indicates the initial value of the purity of the reduced density matrix, i.e., $P_{\mathrm{REF}}=P_{\mathrm{INIT}}$ and the horizontal dashed line shows the initial Schmidt number. Keep in mind that larger values of $P_{\mathrm{REF}}$ mean smaller values of $C_{\mathrm{REF}}$.

## Data Availability

No new data were created or analyzed in this study. Data sharing is not applicable to this article.

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
