# Peer review of "Optimal High-Dimensional Entanglement Concentration for Pure Bipartite Systems"

_micromachines, 2023, doi:10.3390/mi14061207_

Round 1

Reviewer 1 Report

In this manuscript, the authors theoretically derive high-dimensional entanglement concentration in a pure bipartite system and numerically calculate the optimal probabilistic entanglement concentration up to 1024 dimensions. Due to the low success probability of concentration in high dimensions, the authors considered the tradeoff between the final entanglement and the success probability. Then numerical optimization is carried out for the success probability and entanglement amount respectively, to ensure the optimal entanglement concentration under given conditions. The manuscript has clear logic, detailed derivation, and correct conclusions. I recommend publishing it on Micromachines, but before that I would call upon the authors to address the following concerns.

1. In the title, the authors mentioned the optimal high-dimensional entanglement concentration. However, in the main text, the optimization problem is given for the defined concentration efficiency Q (z) (or purity of the reduced density matrix P(y) ). Although Q (z) is a good description of the efficiency of entanglement concentration, can it indicate that the selection of Q (z) is optimal? Or is Q (z) a better choice?

2. In experiments, quantum states are inevitably affected by environmental noise leading to decoherence, while entanglement concentration mainly considers the case of pure states. Therefore, it is important to demonstrate practicality and efficient entanglement concentration. In recent years, there have been many studies on entanglement purification, which is similar to entanglement concentration. I suggest the authors add references to illustrate the experimental progress in this area and provide assistance for future entanglement concentration experiments.

E.g Entanglement purification (distillation) using different degrees of freedom of photons.

[1] Hu X M, Huang C X, Sheng Y B, et al. Long-distance entanglement purification for quantum communication[J]. Physical review letters, 2021, 126(1): 010503.

[2] Ecker S, Sohr P, Bulla L, et al. Experimental single-copy entanglement distillation[J]. Physical Review Letters, 2021, 127(4): 040506.

Entanglement purification of a single degree of freedom

[3] Ecker S, Sohr P, Bulla L, et al. Remotely establishing polarization entanglement over noisy polarization channels[J]. Physical Review Applied, 2022, 17(3): 034009.

[4] Huang C X, Hu X M, Liu B H, et al. Experimental one-step deterministic polarization entanglement purification[J]. Science Bulletin, 2022, 67(6): 593-597.

Author Response

In this manuscript, the authors theoretically derive high-dimensional entanglement concentration in a pure bipartite system and numerically calculate the optimal probabilistic entanglement concentration up to 1024 dimensions. Due to the low success probability of concentration in high dimensions, the authors considered the tradeoff between the final entanglement and the success probability. Then numerical optimization is carried out for the success probability and entanglement amount respectively, to ensure the optimal entanglement concentration under given conditions. The manuscript has clear logic, detailed derivation, and correct conclusions. I recommend publishing it on Micromachines, but before that I would call upon the authors to address the following concerns.

We thank the referee for the time invested in reviewing our manuscript. Her/his concerns will be addressed as follows.

Point 1: In the title, the authors mentioned the optimal high-dimensional entanglement concentration. However, in the main text, the optimization problem is given for the defined concentration efficiency Q (z) (or purity of the reduced density matrix P(y) ). Although Q (z) is a good description of the efficiency of entanglement concentration, can it indicate that the selection of Q (z) is optimal? Or is Q (z) a better choice?

Response 1: The referee is correct in pointing out that efficiency has been defined in terms of a single function and his/her question about whether Q(z) is the optimal choice is legit. From the mathematical point of view, every optimization problem is linked to a function to be optimized. The nature of such a function is given by the main goal of the problem. For example, a company may want to maximize profits and a function f1(x) will be used. But if the same company intends to maximize profits provided additional conditions have to be fulfilled, a different function f2(x) has to be used. In principle, as the requirements for each case are different, we cannot assess which one, f1(x) or  f2(x), is the optimal function of merit that shall be used. It all depends on the context.

Having said that, we proposed Q(z) in such a way that: 

  • Its optimization suggests performing a concentration procedure, in contrast to other figures of merit (e.g., overall mean entanglement) which suggest doing nothing is the best option. For that reason, Q(z) was written using as a difference between a reference entanglement value and the final entanglement value.
  • It contains a user-defined parameter that allows for flexibility in the main goal of the optimization, in contrast to some schemes which only optimize success probability since the resulting entanglement has been fixed to be maximal. 
  • Its optimization can be algebraically analyzed. There are other entanglement measures (e.g., entanglement of formation, conditional entropy, etc.) that are widespread, but very difficult to work with. 

In summary, we cannot say we are proposing the optimal function Q(z) and we cannot either assure there exists an optimal one, but we have proposed a way of thinking about the problem of entanglement concentration in terms of a more flexible optimization. Of course, other researchers may propose other functions to deal with based on their own goals to be fulfilled in the concentration protocol. 

Point 2: In experiments, quantum states are inevitably affected by environmental noise leading to decoherence, while entanglement concentration mainly considers the case of pure states. Therefore, it is important to demonstrate practicality and efficient entanglement concentration. In recent years, there have been many studies on entanglement purification, which is similar to entanglement concentration. I suggest the authors add references to illustrate the experimental progress in this area and provide assistance for future entanglement concentration experiments.

E.g Entanglement purification (distillation) using different degrees of freedom of photons.

[1] Hu X M, Huang C X, Sheng Y B, et al. Long-distance entanglement purification for quantum communication[J]. Physical review letters, 2021, 126(1): 010503.

[2] Ecker S, Sohr P, Bulla L, et al. Experimental single-copy entanglement distillation[J]. Physical Review Letters, 2021, 127(4): 040506.

Entanglement purification of a single degree of freedom

[3] Ecker S, Sohr P, Bulla L, et al. Remotely establishing polarization entanglement over noisy polarization channels[J]. Physical Review Applied, 2022, 17(3): 034009.

[4] Huang C X, Hu X M, Liu B H, et al. Experimental one-step deterministic polarization entanglement purification[J]. Science Bulletin, 2022, 67(6): 593-597.

Response 2: We thank the referee for the bibliographic suggestions. They have been added to the revised version of the manuscript as they truly highlight recent advances in the field. 

Reviewer 2 Report

The manuscript “Optimal High-Dimensional Entanglement Concentration for Pure Bipartite Systems” concerns the study of the quantum entanglement: the authors propose the optimized algorithms to increase the entanglement for a bipartite quantum system of very large dimensions. Specifically, the authors consider a state of the initial two-qudit (D dimensional) quantum system in the Schmidt decomposition. Combined with an ancillary qubit, the system is subjected to a unitary evolution that results in a Kraus mapping of the initial state. The general subject of the article is to develop a procedure for finding optimal parameters of this Kraus mapping that lead to the rising of entanglement with a reasonable probability.

To deal with it, the authors introduce a measure of concentration efficiency Q, which is just the squared product of the probability of success for the desired outcome of the ancillary qubit measurement and the I-Concurrence parameter. To make the probability of the whole procedure reasonable (inevitably at the expense of limiting the entanglement rise), the latter parameter is suggested to shift by some reference value for the best choice of which the arguments are also given. Finally, the main aim is formulated as a quadratic optimization problem intended to attain a) the desired entanglement rise or b) the success probability. The authors elaborate detailed algorithms to solve both cases.

The only point I could ask the authors to clarify is whether Q is guaranteed to have positive values at all if Cref > Cinit: the negative Q in this case means that we failed to reach Cref.

In summary, the article is written thoroughly and is certainly of interest to researchers in the field of quantum information. I recommend it for publication.

Author Response

The manuscript “Optimal High-Dimensional Entanglement Concentration for Pure Bipartite Systems” concerns the study of the quantum entanglement: the authors propose the optimized algorithms to increase the entanglement for a bipartite quantum system of very large dimensions. Specifically, the authors consider a state of the initial two-qudit (D dimensional) quantum system in the Schmidt decomposition. Combined with an ancillary qubit, the system is subjected to a unitary evolution that results in a Kraus mapping of the initial state. The general subject of the article is to develop a procedure for finding optimal parameters of this Kraus mapping that lead to the rising of entanglement with a reasonable probability.

To deal with it, the authors introduce a measure of concentration efficiency Q, which is just the squared product of the probability of success for the desired outcome of the ancillary qubit measurement and the I-Concurrence parameter. To make the probability of the whole procedure reasonable (inevitably at the expense of limiting the entanglement rise), the latter parameter is suggested to shift by some reference value for the best choice of which the arguments are also given. Finally, the main aim is formulated as a quadratic optimization problem intended to attain a) the desired entanglement rise or b) the success probability. The authors elaborate detailed algorithms to solve both cases.

The only point I could ask the authors to clarify is whether Q is guaranteed to have positive values at all if Cref > Cinit: the negative Q in this case means that we failed to reach Cref.

In summary, the article is written thoroughly and is certainly of interest to researchers in the field of quantum information. I recommend it for publication.

Response: We thank the reviewer for her/his kind review and we shall proceed to answer his/her question about the positivity of Q(z). 

The referee is correct in saying that Cref>Cinit means we failed at the attempt of concentrating entanglement beyond the chosen reference value. But optimization will always lead to a nonnegative value. The most extreme case will be if Cref = Cmax, which means the reference value has been set as the maximum attainable value the entanglement may adopt (i.e., the standard entanglement concentration procedure). In such a case, Q(z) will be always negative unless the resulting entanglement is exactly equal to Cmax, which is actually the solution we have found in many numerical and algebraic simulations. Therefore, although Q(z) may adopt negative values, the optimization will always yield a positive value.